# Emerging Potential of Exosomes in Regenerative Medicine for Temporomandibular Joint Osteoarthritis

**DOI:** 10.3390/ijms21041541

**Published:** 2020-02-24

**Authors:** Yeon-Hee Lee, Hee-Kyung Park, Q-Schick Auh, Haram Nah, Jae Seo Lee, Ho-Jin Moon, Dong Nyoung Heo, In San Kim, Il Keun Kwon

**Affiliations:** 1Department of Orofacial Pain and Oral Medicine, Kyung Hee University Dental Hospital, #26 Kyunghee-daero, Dongdaemun-gu, Seoul 02447, Korea; omod0209@gmail.com (Y.-H.L.); dental21@khu.ac.kr (Q.-S.A.); 2Department of Oral Medicine and Oral Diagnosis, Dental Research Institute, Seoul National University School of Dentistry, Seoul 03080, Korea; dentopark@snu.ac.kr; 3Department of Dentistry, Graduate School, Kyung Hee University, Seoul 02447, Korea; hrnah@khu.ac.kr (H.N.); leejaeseo@khu.ac.kr (J.S.L.); 4Department of Dental Materials, School of Dentistry, Kyung Hee University, Seoul 02447, Korea; 3216@khu.ac.kr (H.-J.M.); dnheo81@gmail.com (D.N.H.); 5Center for Theragnosis, Biomedical Research Institute, Korea Institute of Science and Technology (KIST), Seoul 02792, Korea; iskim14@kist.re.kr

**Keywords:** temporomandibular joint, osteoarthritis, exosome, regenerative medicine, mesenchymal stem cell, osteochondral regeneration

## Abstract

Exosomes are nanosized vesicles (30–140 nm) of endocytic origin that play important roles in regenerative medicine. They are derived from cell membranes during endocytic internalization and stabilize in biological fluids such as blood and synovia. Temporomandibular joint osteoarthritis (TMJ OA) is a degenerative disease, which, in addition to chronic pain, is characterized by progressive cartilage breakdown, condylar bone remodeling, and synovitis. However, traditional clinical treatments have limited symptom- and structure-modifying effects to restore damaged cartilage and other TMJ tissues. This is due to the limited self-healing capacity of condylar cartilage. Recently, stem-cell-derived exosomes have been studied as an alternative therapeutic approach to tissue repair and regeneration. It is known that trophic regulation of mesenchymal stem cells (MSCs) has anti-inflammatory and immunomodulatory effects under pathological conditions, and research on MSC-derived exosomes is rapidly accumulating. MSC-derived exosomes mimic the major therapeutic effects of MSCs. They affect the activity of immune effector cells and possess multilineage differentiation potential, including chondrogenic and osteogenic differentiation. Furthermore, exosomes are capable of regenerating cartilage or osseous compartments and restoring injured tissues and can treat dysfunction and pain caused by TMJ OA. In this review, we looked at the uniqueness of TMJ, the pathogenesis of TMJ OA, and the potential role of MSC-derived exosomes for TMJ cartilage and bone regeneration.

## 1. Introduction

The temporomandibular joint (TMJ) is a unique joint that connects the mandibular condyle with the articular surface of the temporal bone, accompanied by hinge and gliding activity. It is one of the most frequently used and most complex joints in the human body [1]. Osteoarthritis (OA) is a progressive degenerative disorder primarily affecting the joints and is characterized by cartilage degradation, destruction of subchondral bone, osteophyte formation, synovitis, muscle soreness, and chronic pain [2]. Osteoarthritis of TMJ (TMJ OA) often involves degeneration of both hard and soft tissues of TMJ, and patients with TMJ OA commonly have joint pain and dysfunction with a deterioration in quality of life. 

Approximately 15% of the global population suffers from OA [3]. The prevalence and symptom rates of TMJ OA are similar to generalized OA. In TMJ OA, based on clinical symptoms of the disease, it affects 8%-16% of the population. [4,5]. Temporomandibular disorders (TMDs) in women are about twice more prevalent than in men, and 80% of patients who seek treatment for TMD are women [6]. Furthermore, unlike OA of other joints, the prevalence is highest among women aged 18 to 45 years [6]. Moreover, women have increased susceptibility to the initiation of TMJ OA and induced pain, occurring mainly after puberty during the reproductive years [7,8]. TMJ OA is usually characterized by secondary articular disc displacement, trauma, parafunctional habit, and functional overloading [9]. However, the pathology of TMJ OA is complex; the TMJ and its surrounding structures are involved in multifactorial traits and processes [10]. Furthermore, the pathogenesis and underlying molecular mechanisms involved in TMJ OA development remain veiled and unknown. 

Because of the limited self-healing ability of articular cartilage, TMJ OA is one of the most difficult joint diseases, and currently there is no consensus treatment for complete remission. The treatment of OA is aimed at pain relief and improvement of affected joint function. The therapeutic strategy of TMJ OA aims to prevent gradual destruction of cartilage and subchondral bone, induce bone remodeling, relieve joint pain, and restore TMJ functions [11]. Generally, applied pharmacologic therapy shows efficacy in pain relief but is frequently associated with adverse effects. Traditional clinical treatments mainly include nonsurgical options, such as psychotherapy, physical therapy, occlusal stabilization splints, medication, and arthrocentesis, while surgical intervention has been applied to patients with severe symptoms [11,12]. Although these abovementioned treatments can prevent disease progression to a certain degree, they are unable to completely restore degraded cartilage or subchondral bone lesions, as well as disc deteriorations. Articular cartilage lesions often lead to severe symptoms, such as activity-related pain, swelling, and dysfunction, and if not treated properly, are a major risk factor for OA. [11]. In other words, the deterioration and persistence of these lesions results in friction between the bones of the mandibular condyle and the temporal articular surface during TMJ movement, which leads to further progression to TMJ OA.

Mesenchymal stem cells (MSCs) have great potential for cell therapy for articular cartilage repair, which can be affected by MSC-derived exosomes. The efficacy of autologous or allogenic MSCs in cartilage repair has been demonstrated in animal studies [10,13], and more recently in human clinical trials [14]. The use of MSCs to repair cartilage tissue was predicated on the hypothesis that these cells could differentiate into chondrocytes to replace damaged tissue [15]. However, in recent years, increasing evidence suggests that MSCs secrete a wide range of trophic factors to modulate the injured tissue environment and to orchestrate subsequent regenerative processes, including cell migration, proliferation, differentiation, and matrix synthesis [16]. In addition, increasing evidence demonstrates that the positive effects of such MSC-based therapy are mediated by exosomes released from administered cells and that their exosomal cargo is largely responsible for therapeutic effects. In summary, accumulative evidence has shown that MSCs exert a therapeutic impact through paracrine-mediated effects but not by direct cell replacement [17,18]. In the last decade, MSC exosomes were found to be effective against several MSC disease targets and were reported to mediate cartilage repair and regeneration [17,18]. 

In this review, we discuss the characteristics and properties of MSCs, the current understanding of MSC exosomes, and advances in our knowledge of their emerging role in treating TMJ OA. Given the limited understanding of its pathogenesis and the low healing potential of avascular cartilage, no effective therapy is available for restoring TMJ structures with progressive OA. Here, we present novel perspectives for the development and implementation of MSC exosomes as a cell-free regenerative medicinal approach for cartilage repair and regeneration of TMJ. For this purpose, four of the authors of this review conducted a literature search. For a period of 32 years prior to January 2020, we searched the literature via PubMed and Google scholar search engines, and we selected the top 99 articles that matched the theme.

## 2. TMJ and cartilage of TMJ

### 2.1. Temporomandibular Joint (TMJ)

The TMJ is a complex articulation composed of mandibular condyle and articular surfaces of the temporal bone. Both the temporal articular surface and mandibular condyle are covered by dense articular fibrocartilage. The temporal articular fossa is composed of the mandibular fossa and articular tubercle (Figure 1) [19]. Along the long joint on the temporal surface, each TMJ on the right and left sides has a wide range of motion consisting of rotation and translation, creating a bodily movement of the mandible. The dense fibrocartilaginous disc can relieve pain from mechanical stress which occurs between the condylar surfaces and temporal articular surfaces. TMJ discs, located between the superior and inferior joint spaces, have a high collagen content for durability and rigidity. Uniquely, TMJ discs do not have direct nerve distribution or vascularization due to their complex anatomic features. Alternatively, retrodiscal tissue, which is a posterior attachment of TMJ disc, is characterized by many blood vessels and nerves that are crucial in pathophysiological processes [20].

### 2.2. Uniqueness of TMJ

TMJ is one of the most unique human joints in terms of composition and development. The TMJ in humans consists of two joints in one bone. The two joints, respectively, connect the head and the mandible. One joint may influence the function of the other. In other synovial joints of the body, the joint surface is covered by a layer of hyaline cartilage [21]. However, the TMJ is distinct and special in the fact that it is composed of fibrocartilage [22]. A unique property of fibrocartilage is that it contains both collagen types I and II, compared with articular hyaline cartilage which contains only type II collagen [23]. Thus, fibrocartilage can withstand shear forces better than hyaline cartilage, because it contains excellent tissue components that can withstand the large amount of occlusal loads applied on the TMJ [24]. Another strength of fibrous cartilage of the TMJ relative to hyaline cartilage is the high density of fibers that can withstand static load and forces of movement, which are less likely to break down over time and less affected by aging than are other joints [24]. 

In the developmental stages of the TMJ, MSCs split within themselves to divide into even smaller cells, eventually attaining full size. Then, these MSCs migrate into the interior condyle and then into the cartilage, where cell differentiation occurs, and then the differentiated cells become immature chondrocytes [21]. The growth in the TMJ cartilage occurs mainly through the differentiation of mesenchymal tissue, rather than under the influence of mitosis of cartilage progenitor cells. The TMJ is not fully characterized but shows histological, anatomical, and developmental differences from other body joints, which are likely to affect the cause, predisposition, or progression of TMJ disease and require further study.

## 3. Pathogenesis of TMJ OA

TMJ OA, a degenerative TMJ disease, occurs due to an imbalance in the synthesis and degradation of the condylar matrix. Degenerative changes in the TMJ are mediated by chondrocytes and fibrocartilage cells in the fibrocartilage of TMJ. This results in a gradual loss of extracellular matrix (ECM) components of subchondral bone and/or articular cartilage [21]. The articular surface is covered with fibrocartilage to resist mechanical stress, but the underlying subchondral bone is stress sensitive, and hence extensive remodeling occurs. Moreover, cartilage is vulnerable to damage caused by wear and tear over time, since it is responsible for the load-bearing capacity. Usually, structural defects in the cartilage are not spontaneously healed. Insufficient self-healing capacity of cartilage can be caused by a lack of blood supply and reduced metabolic activity. If damaged cartilage is not properly treated, subjective symptoms not only worsen but also adversely affect the surrounding tissue, ultimately turning into OA [25,26]. Microscopically, the loss of collagen and proteoglycans is observed in OA cartilage, resulting in loss of ECM structure, thus impairing the biomechanical properties of the joints [27]. 

Inflammation of the synovial membrane, termed synovitis, and subchondral sclerosis can also be caused by structural and mechanical changes in the OA cartilage, making OA more debilitating [9]. In this process, biomechanical and biochemical changes not only interfere with cartilage homeostasis but also contribute to the development and progression of TMJ OA. This results in narrowing the space of the joint cavity, destroying the cartilage, and clinically causing a decrease in joint function. Each of these etiological factors is discussed in detail below (Figure 2).

### 3.1. Inflammation

TMJ OA is classified as a localized low inflammatory arthritis condition, in contrast to rheumatoid arthritis, which is a systemic high inflammatory condition [11]. Researchers have paid considerable attention to the importance of the inflammatory response in the progression of TMJ OA. In patients with TMJ OA, several inflammatory cytokines, including interleukin (IL)-12, IL-6, IL-1β, and tumor necrosis factor (TNF)-α, increase in synovial fluid [28]. In addition, the levels of monocyte chemoattractant protein (MCP)-1 in patients with TMJ OA is also elevated in inflammatory synovial tissue, which is highly upregulated in IL-1β-stimulated synovial cells [29]. However, potential prognostic indicators or diagnostic markers for TMJ OA have not been identified to date, and further investigation is needed to determine whether the markers for cartilage degradation in TMJ synovial fluid increase.

### 3.2. Excessive Mechanical Stress and Malocclusion

Excessive mechanical stress beyond normal adaptive capacity is considered to be a major factor causing cartilage breakdown in the TMJ [30]. Unilateral TMJ OA is associated with mandibular asymmetry and increased internal muscle activity in the electromyography test of masticatory muscles on the OA side [31]. The effect of mechanical stress on mandibular cartilage and chondrocytes has been investigated both in vitro and in vivo. Due to excessive mechanical stress, activation of the plasminogen activator system is induced, which can lead to proteolysis of ECM components [32]. However, the causal relationship between TMJ OA, facial asymmetry, and masticatory muscle overuse is not yet clear. In a recent study, long-term experimental malocclusion is caused by subchondral bone defects and enhancement of osteoclast activity; as a result, the newly formed subchondral bone had poor bone mineral density and low load-bearing capacity for mechanical stress [33]. There is considerable evidence that abnormal biomechanical or mechanical stimuli play an important role in the initiation and progression of TMJ OA. To establish the TMJ OA model, excessive load on the joints, unbalanced use of bilateral TMJ, and abnormal dental malocclusion were considered [34]. The death of endoplasmic reticulum stress-induced cells dilutes the mandibular cartilage layer, which occurs as part of the process of chondrocyte apoptosis induced by mechanical stress [35]. On the contrary, proper mechanical stress is important, especially for the development of the mandibular condyles. Three-week-old female mice with trimmed incisors and soft diet for four weeks were reported to have thinned cartilage and reduced subchondral bone mass in the TMJs [36].

### 3.3. Apoptosis or Necrosis of Chondrocytes

Cartilage cell death caused by apoptosis or necrosis is considered to be a central feature of degenerative changes in cartilages affected by osteoarthritis. Experimentally, apoptosis of chondrocytes was associated with the onset of early stages of cartilage degradation in rat models of iodoacetate induced TMJ OA, and degradation of cartilage and an increase in cytokines released from apoptotic cartilage were observed [37].

There is also accumulating evidence that chondrocytes play a major role in the destruction of subchondral bone. The upregulation of catabolic enzymes in cartilage matrix, such as matrix metalloproteinases (MMPs), disintegrins, and MMPs with thrombospondin motifs (ADAMTS), is a result of the pathologic changes of TMJ OA [38]. In addition, the upregulation of ADAMTS-5 was observed in condylar cartilage during early stages of TMJ OA [39]. It is known that the molecular mechanism of cartilage destruction involves a process by which catabolic enzymes break down ECM. A previous in vitro study suggests a molecular mechanism with IL-1β-induced catabolism of the condylar matrix due to the upregulation of Wnt-5A, which is activated in the NF-κB signaling pathway [40]. 

### 3.4. Sex Hormonal Effect

TMJ OA is more prevalent in females than males and occurs mainly after puberty, especially during the reproductive periods [41], suggesting that female sex hormones play a role in the course of the disease. Sex-based determinants, such as hormonal effects from the action of estrogen, progesterone, and relaxin, can make individuals vulnerable to TMJ OA [21,41]. There are several levels of evidence to support this hypothesis. Estrogen receptors in TMJ have been shown to have sexual dysplasia. In addition, estrogen and progesterone receptors were found in both men and women in humans, but only in the male population of rats [21,42]. Estrogens inhibit chondrocyte proliferation of the TMJ via an estrogen receptor (ER)-β-dependent mechanism [43]. In addition, elevated levels of estrogen were observed in women with TMJ disease [44]. These findings suggest a potential role of specific sex hormones in causing or predisposing to TMJ degeneration. Moreover, estrogens and relaxins contribute to TMJ degeneration via enhanced expression of histolytic enzymes belonging to the MMP family from TMJ fibrocartilage [45,46]. MMPs can break down the major matrix macromolecules of TMJ cartilage, comprising of collagen and proteoglycans, as well as small proteins in this tissue. However, the direct and precise mechanism by which female reproductive hormones can induce TMJ degeneration has not been discovered until recently [47], and further research is needed.

### 3.5. Genetic Factors

Another possible mechanism by which TMJ OA may arise is from alterations in the ECM; this might be due to genetic disturbances that directly alter the composition of the ECM in the TMJ, or indirectly due to changes in composition and/or turnover of the ECM caused by other genetic factors. Several genes are associated with the painful TMJ disorder as discovered by a genome-wide association study (GWAS) [48]. In a recent GWAS study, 22 independent loci were found that showed an implicit association with degenerative bone changes in the TMJ [49]. To date, several mouse models have been published showing that TMJ OA occurs when ECM of the mandibular cartilage is defective. In one of these models, transgenic mice mutated with genes driven by human collagen type II promoters affect human type II collagen gene expression and allow mutant proteins to be expressed only in cartilage [50]. In mice with this mutation, subchondral cysts developed in the mandibular condyle, and nearly complete resorption of TMJ cartilage was noted. Another mouse model of TMJ OA is the loss-of-function mutation in the gene encoding the alpha1 chain of type XI collagen in *cho* (chondrodysplasia) mice [51]. In addition, heterozygote *cho* mice show evidence of osteoarthritic changes in proteoglycan staining, which are observed in the surface articular layers of the mandibular condylar cartilage. Another TMJ OA mouse model is obtained by disruptions in the production of ECM proteoglycans, such as biglycan and fibromodulin [52]. 

In the early stages of TMJ OA, upregulation of genes involved in osteoclast activity and/or an increase in RANKL/OPG ratio in subchondral bone contributes to an increase in subchondral bone turnover in biglycan/fibromodulin-deficient mice [53]. Osteoblast-specific transforming growth factor (TGF) -β1 transgenic mice in the bone marrow were used to have high levels of active TGF-β1 to assess the effect of overexpressed TGF-β1 on TMJ OA [54]. Through this model, excessive apoptosis of the mandibular chondrocytes, upregulation of MMP-9, MMP-13 and vascular endothelial growth factor (VEGF) in chondrocytes, and decreased bone mineral density were observed. This suggests that TGF-β1 plays a crucial role in increasing subchondral bone turnover in early stages of TMJ OA.

These animal models point out interesting possibilities for the genetic cause of degenerative TMJ disease. However, there is currently no clear evidence of the effect of these genetic factors on human TMJ OA. Of course, there is a strong possibility that future research will find an association between TMJ OA and genetic defects that undermine the robustness of sECM. This may also provide information to identify individuals predisposed to developing TMJ OA.

## 4. Current Status of TMJ OA Treatments

### Overview of TMJ OA Treatment 

The etiology of TMJ disorders is multi-factorial and complex, and the pathophysiology of TMJ OA is still poorly understood. Etiologic factors may include macrotrauma, parafunctional habit, bruxism, malocclusion, estrogen influence, genetic variations, and even psychological problems [11]. One or more factors contribute to TMJ OA in individuals in various combinations, but fundamentally, mechanical overload beyond the physiological resistance to joint components leads to TMJ OA. The best treatment strategy for TMJ OA is to identify its main causes and eliminate them [11]. The current clinical management options of TMJ OA include noninvasive, minimally invasive, and invasive techniques (Figure 3). 

Such treatments can be used in various combinations and are applied to eliminate the potential cause of TMJ OA and to treat the symptoms. Examples of noninvasive treatments are occlusal stabilization splints, medications, and physical therapy. The use of occlusal stabilization splints has been used since the 18th century, and they are still commonly used [55]. Noninvasive medications consist of corticosteroids, nonsteroidal anti-inflammatory drugs (NSAIDs), muscle relaxants, opioids, anxiolytics, antidepressants, anticonvulsants, and benzodiazepines [56]. Traditionally, many drugs have been used for TMJ OA, but related unsatisfactory and unacceptable side effects have often been reported. Thus, there is ongoing research for potential new drugs. Based on the therapeutic goals of TMJ OA, anti-inflammatory cytokines matrix degradation inhibitors, chondrogenesis inducers, apoptosis inhibitors, and osteogenesis inhibitors have recently been studied [57]. Similarly, reports of long-term effects or side effects for these latest oral or topical drugs continue.

Minimally invasive therapies are as widely used as medication, targeting masticatory muscles or the TMJ itself. Botulinum toxin (Botox) has little side effects and has been injected into the masticatory muscle, including the masseter, temporal, and lateral pterygoid muscles [58]. Minimally invasive treatment of the TMJ itself includes intra-articular injection, arthroscopy, and arthroscopy. Mostly intra-articular injection of two components, hyaluronic acid and corticosteroids, is used selectively for the treatment of TMJ OA [59,60]. Intra-articular injection of hyaluronic acid into the affected joint compartments improves TMJ OA symptoms. Hyaluronic acid is a normal component of joint fluid in the TMJ of healthy people. Corticosteroid injections are recommended for patients with moderate to severe pain who do not respond to oral analgesic and anti-inflammatory drugs [61]. Arthrocentesis usually means a lavage of the superior or inferior TMJ space using saline solution. The pressure generated during irrigation may eliminate adhesions [62]. In addition, the miniaturization of endoscopes allowed arthroscopic examination of the TMJ areas. Endoscopy enables proper visualization, manipulation, and removal of pathological intra-articular tissue. Small-diameter instruments allow direct elimination of pathogenesis in the TMJ, with high success rates [20]. The short- and long-term effects of these minimally invasive treatments have been well reported.

In general, invasive treatments are only performed for patients suffering severely from osteoporosis, neoplastic, and developmental disorders in the TMJ [10]. Invasive treatment of the TMJ consists of open joint surgery aimed at restoring joint tissue or completely replacing the TMJ with an autogenous or allogeneic material. Functional discs are critical to the success of mandibular condyle regeneration in vivo, and various measures have been attempted to replace the TMJ disc [10]. However, to date, some of the treatments proposed for disc replacement did not produce satisfactory clinical results. Thus, TMJ regeneration strategies are being rethought more carefully. The limitations of current therapeutics for TMJ OA have naturally increased interest in regenerative strategies that combine cells, well-targeted biologically active molecules, and implantable scaffolds. Recent achievements in regenerative medicine related to joint problems can contribute to meeting this unique and complex regenerative challenge.

## 5. MSC Exosomes in Joint Diseases

### 5.1. Exosomes 

Exosomes are a type of nanosized extracellular vesicles (EV) with a diameter of 30 to 140 nm. After the fusion of multi-vesicular bodies (MVB) with cell membranes, it is secreted into the extracellular space in most cell types [63]. Exosomes were first reported by Pan and Johnstone in the early 1980s and were considered as cellular waste products of erythrocytes during sheep reticulocyte maturation [64]. Exosomes are secreted from many cell types, including lymphocytes comprising B and T lymphocytes, platelets, mast cells, dendritic cells, and tumor cells. Secreted exosomes are found in most body fluids such as blood, urine, breast milk, cerebrospinal fluid, and saliva (Figure 4). 

The major role of exosomes is considered to be as a vehicle for intercellular communications and to transfer cargo contents to recipient cells. Exosomes are essential for cell-to-cell communication and function to induce biological responses in recipient cells by delivering proteins, lipids, and nucleic acids (mRNA and miRNA) from the parent cell [65]. Exosomes play an important role in cellular communication by transporting their biological cargo in OA areas and can induce bone mineralization and thickening and lead to loss of stiffness [66,67]. The cargo contents of the exosomes are specific to the cell of origin and can transmit parent cell signals to neighboring or target cells without direct cell-to-cell contact. Exosomes also contribute to the maintenance of homeostasis and the healing of disease, through tissue–tissue and cell–cell communication. Regardless of the parent cell, there are also components that exosomes have in common. Exosomes contain common elements, such as heat shock proteins (Hsp 60, Hsp 70, and Hsp 90), tetrasparin (CD9, CD63, and CD81), nucleic acids (mRNA, miRNA, long noncoding RNAs, and DNAs), membrane transport and fusion proteins (GTPases, annexins, and Rab proteins), biosynthesis-related proteins (Alix and TSG 101), and lipids (cholesterol and ceramide) [68,69]. Based on these unique properties of exosomes, new and innovative approaches to diagnosis and treatment are possible. Studies on cargo contents, biological production, and release mechanisms of exosomes allow researchers and clinicians to better target the treatment of certain diseases. In terms of drug delivery, researchers can use exosomes as a natural drug delivery vehicle to increase the targeting accuracy of exosomes, minimize the dosage of drugs, and reduce the side effects. However, despite substantial efforts in a relatively new field of research, our understanding of the treatment of TMJ OA with exosomes is limited. Inefficient separation methods, lack of high-resolution visualization techniques, and absence of exclusive biomarkers need to be addressed, so clinical application of exosomes is expected to take more time. 

### 5.2. MSC Exosomes in Therapeutics 

Until recently, MSCs (mesenchymal stem cells), including MSCs derived from adipose and bone marrow tissues, have been used as representatives of cell-based therapies. Mechanisms regarding the therapeutic potential of MSCs are based on (1) transfer of exosomes, with a relatively narrow range of sizes (diameter of 30 - 140 nm) or microvesicles with a broad size distribution (150–1000 nm), and (2) the action of paracrine factors, including peptides, proteins, and hormones, have been proposed [70]. 

Increasing experimental/clinical evidence indicates that MSC-derived exosomes (MSC exosomes) might become the new cell-free therapy agents with attractive advantages over MSCs, such as no risk of tumor formation and low immunogenicity, and they also play a crucial role in enhancing angiogenesis, which is fundamental for tissue repair [71]. The cargo of MSC exosomes functions through delivery to a recipient cell containing more than 150 different miRNAs and more than 850 unique proteins inside and is involved in activity of target cells through various pathways [72,73] (Table 1). 

The mechanism by which MSCs facilitate tissue repair is through the release of bioactive factors, such as exosomes in which MSCs are trophic, immunomodulatory, and/or regenerating at the site of injury or disease [71]. Recently, several methods using exosomes, described in detail below, have approached the regeneration and treatment of damaged tissues (Figure 5). After exosome administration in animal or human models, current knowledge of their dynamic biodistribution or degradation/excretion and of their safety is limited. However, according to Lai et al., biodistribution proceeds in stages of liver and lungs for 30 min after exosome administration, and exosomes are removed within 1 to 6 h after administration via liver and kidney treatment [74]. In addition, according to Grange et al., exosomes accumulate in damaged tissues rather than in healthy tissues [75]. Thus, it can be hypothesized that MSC-derived exosomes have favorable biodistribution and pharmacokinetic profiles.

### 5.3. Potential Roles of MSC Exosomes in TMJ Regeneration

MSC exosomes are expected to promote the regeneration of damaged TMJ, with a complex structure (Figure 6). MSC exosomes enhance matrix synthesis consisting of type II collagen and sulfated glycosaminoglycans (s-GAG), accelerating neoplastic tissue filling. In mice treated with exosomes at the site of cartilage defects, almost complete recovery of cartilage and subchondral bone was observed, characterized by regular filling of hyaluronic cartilage on the surface, complete binding to adjacent cartilage, and recovery of the ECM layer. In contrast, in saline-treated controls, only fibrous repair was observed at the site of the defect. [76]. The mechanism of cartilage regeneration by the MSC exosomes is not clear since the mechanisms of the other therapeutic effects of the MSC exosomes are inaccurate. 

MSC exosome, the vehicle responsible for intercellular communication, is well suited for delivering timely MSC responses in an effective manner to restore homeostasis of the microenvironment. The bi-lipid membrane structure of MSC exosomes plays an important role in endocytosis or membrane fusion within cells and is important because it provides multifunctionality for interacting with various cell types through extracellular membrane receptor–ligand interactions. Moreover, MSC exosomes are rich in proteins and enzymes, which can regulate and restore homeostasis of ECM [77]. 

Most importantly, the cargo of MSC exosomes is rich in enzymes and proteins, and these factors have a unique ability to restore homeostasis. Enzyme-mediated activity is a catalytic reaction. The activity of the enzyme is increased in proportion to the impairment of the homeostatic equilibrium between the enzyme substrate and the product. When homeostasis is impaired during injury and disease, enzyme-based exosomes are activated. Activated exosomes restore homeostasis and promote tissue function, as well as contribute to tissue repair and regenerations. When homeostasis is restored and tissue damage is healed, exosome enzyme activity is also reduced. Thus, exosome-based therapeutics are very sensitive to the disease precipitating microenvironment, which is also a limitation [78]. In addition, all MSC exosomes have a common set of RNAs, lipids, and proteins. Although exosomes are released from various cells, they may have similar therapeutic activity because they share evolutionarily conserved RNA, lipid, and protein sets [79]. 

MSC exosomes have advantages over exosomes from other cell types. This is due to the low immunological response, high clinical safety, ease of extraction from a wide range of tissues, and high potential for therapeutic extensibility. The set of evolutionarily conserved cargoes in exosomes contains many housekeeping enzymes involved in the regulation of bio homeostasis, energy conservation, cell number maintenance, and immune function [77]. We would like to propose the possibility that MSC exosomes may promote cartilage repair and regeneration with endogenous endurance by restoring homeostasis in OA-affected areas. The potential mechanism presented following suggests that MSC exosomes can have a regenerative effect on TMJ OA (Figure 7).

#### 5.3.1. Restoring Bioenergetic Homeostasis

Mitochondrial damage and dysfunction is related to the pathogenesis of OA. Chondrocytes in the area where OA occurred have a decrease in mitochondrial biological production and a decrease in mitochondrial electron transport chain (ETC) protein [80,81,82]. ETC is essential for the production of adenosine triphosphate (ATP) by oxidative phosphorylation in mitochondria. The activity of ETC and a decrease in the number of mitochondria will lead to inefficient ATP production and loss of energy homeostasis.

This loss of energy is associated with a series of processes observed in damaged cells in the OA state. Specifically, it causes increased oxidative stress, inflammation of cytokine-induced chondrocytes, disturbed matrix biosynthesis of chondrocytes, increased reactivity of growth factors, increased chondrocyte death, and catabolism and calcification of the cartilage matrix layer [80,81]. In such a scenario, the repair capacity of OA chondrocytes is severely impaired. This is because maintenance of bioenergy homeostasis is essential for initiating and restoring regenerative activity in OA chondrocytes. Importantly, the MSC exosomes carry a cargo copious in active sugar-degradable ATP producing enzymes such as pyruvate kinase, and ATP producing enzymes such as nucleoside-diphosphate kinase and adenylate kinase. Possible mechanisms of action of MSC exosomes are: the glycolytic- and ATP-producing enzymes in exosome cargo promote the increase of ATP level enzymes and compensate for reduced mitochondrial ATP production in OA chondrocytes. Thus, MSC exosomes protect the energy homeostasis in the repair and regeneration of damaged cartilage [82]. ATP synthesis per glucose molecule is inefficient in glycolysis compared to ATP produced by mitochondrial oxidative phosphorylation. However, the cell capacity to increase the glycolytic breakdown flux of 10–100 times easily compensates for this inefficiency. In addition, the corresponding degradation process produces metabolic intermediates for the process of protein assimilation. In addition, it helps to restore the redox potential needed to facilitate the recovery of OA cartilage.

#### 5.3.2. Recovery of Cell Numbers

Damaged cartilage is generally aggravated by inflammatory reactions, which in turn leads to cell death, matrix degradation, structural changes in OA, and loss of function [83]. Cells in the OA region die predominantly through apoptosis induced as a result of inflammatory responses, mitochondrial dysfunction, and oxidative stress present during OA.

The proper number of cells is important for maintaining tissue structure and restoring impaired function. The reduced number of cells should be increased to a range that can maintain steady-state levels of homeostasis. To enable matrix tissue formation and deposition at the defective cartilage site, mesenchymal cell proliferation is essential after initiating the cartilage repair response [84]. In injured cartilage tissue, MSC exosomes can induce proliferation of cells through adenosine-mediated phosphorylation of kinases. When a tissue is damaged by a trauma from mechanical loading, shear stress, chemical toxin, or hypoxia, cells emit dangerous signals, such as ATP [85,86]. Increased extracellular ATP stimulates immunogenic cells to remove damaged or dying cells [87]. However, it is a problem if the damage to the tissue is not resolved and persists. Sustained tissue damage results in excessive ATP signal-induced cell death and a “bystander effect” on healthy neighboring cells. This can lead to a net loss of cell numbers and impaired tissue function. For tissue repair, cell proliferation should be initiated with appropriate reduction in extracellular ATP death signals. By the catalytic action of the highly regulated enzyme CD73, extracellular AMP is hydrolyzed to adenosine. Thus, the death signal presented by extracellularly increased ATP can be easily converted to adenosine by CD73, which can be turned into a prosurvival signal [88]. 

This CD73 action is a characteristic of MSC exosomes. MSC exosomes activate the phosphorylation of survival kinases ERK and AKT in the presence of AMP via the theophylline-sensitive adenosine receptor [89]. Thus, CD73 possessed by MSC exosomes can restore prodeath ATP and prosurvival adenosine signaling to equilibrium. Thus, MSC exosomes can convert ATP death signals from damaged tissue to adenosine presurvival signal via CD73 to initiate cell proliferation for tissue regeneration (Figure 8).

#### 5.3.3. Immunomodulatory Activity

The immune system is essential for the survival of all organisms. It is primarily responsible for defending the body from harmful pathogens and the external environment. In recent years, there are studies that show that the immune system has an important effect on tissue repair. Rapid upregulation of proinflammatory cytokines, including IL-1, IL-6, IL-8, and MMPs, at damaged cartilage sites plays a role in the degradation of the matrix layer and joint damage [82]. These proinflammatory cytokines and MMPs are produced not only by synovial cells but also mainly by immune cells, such as macrophages, and are involved in the onset and progression of OA [83]. 

M1 macrophage polarization in OA synovial tissue inhibits chondrogenic differentiation of MSCs via IL-6 in vitro. On the other hand, M2 macrophage polarization has recently been reported to produce anti-inflammatory IL-10, thereby inhibiting harmful inflammation [90,91]. Therefore, when performing cartilage regeneration therapy at the damaged cartilage or TMJ OA site, the control of proinflammatory environment by macrophages will be important.

The immunomodulatory activity of MSC exosomes requires multifactor synergy, rather than the action of a single factor. Co-operation and synergy of secreted factors, such as IL-6, interferon (IFN)-γ, TGF-β1, prostaglandin E2 (PGE2), hepatocyte growth factor (HGF), and hemeoxygenase-1 is required [92]. MSC exosomes with proteome of >200 immunomodulatory proteins are the ideal mediators for this synergy [73,77]. MSC exosomes also induce high levels of attenuated proinflammatory IL-1, IL-6, TNF-α, and IL-12, and anti-inflammatory factors, IL-10 and TGF-1, in vitro. [93]. In mouse models, injection of MSC exosomes into the allogeneic skin grafts of mice with active immune reactivity induced regulatory T cell production, thereby enhancing the survival of allogeneic skin grafts [93]. 

In addition, MSC exosomes have been reported to modulate various inflammatory responses due to myocardial injury, hypoxic pulmonary hypertension, diabetic skin wounds, and retinal damage [82]. For example, in mouse models of hypoxic pulmonary hypertension, MSC exosomes have been shown to inhibit the influx of macrophages and to induce proinflammatory MCP-1, thus preventing disease progression [94]. The immunomodulatory role of MSC exosomes has not yet been addressed in OA. However, exosomes contained in serum play a crucial role in protecting OA cartilage from the destruction of s-GAG in the presence of proinflammatory IL-1. In addition, the increase of M2 macrophages in the treatment with MSC exosomes was observed to promote cartilage regeneration, suggesting that MSC exosomes can alleviate OA status.

## 6. MSC Exosomes: Next-Generation Therapeutics for TMJ OA

The high prevalence of TMJ OA is the reason we should be alert, and when the symptoms of TMJ OA are severe, clinicians should consider ways to achieve TMJ regeneration. Recently, the development of regenerative medicine in the field of orthopedics can provide the answer to the problem of TMJ regeneration. However, owing to the uniqueness and complexity of the TMJ, anatomical, structural, and functional regeneration is not easy and is very challenging to tackle. The ligaments form a tight link between the mandibular condyle with fibrocartilaginous properties and the dense fibrous disk, which usually starts and progresses when the rigidity of this relationship is compromised. The difficulty in engineering/technical regeneration of the TMJ is not in obtaining pure hyaline cartilage from the basal bone but in obtaining long-term fibrocartilage that has been separated from the basal bone without fibrous or bony adhesion or ossification. In tissue engineering approaches to treat cartilage lesions and OA in animal and human studies, therapies for use with scaffolds have been intensively investigated [95,96]. The use of exosomes derived from chondrogenic and/or osteogenic cells in biomaterial scaffolds could provide a new cell-free therapeutic paradigm for TMJ tissue regeneration [96]. TMJ regeneration using MSC exosomes will require generating scaffolds to mount cells or exosomes and active molecules, or other regulators to mount cells or exosomes, and research to obtain these scaffolds will continue. Numerous therapeutic strategies have been proposed, focusing on the recovery of function and restoring structure of the joint using MSC exosomes (Figure 9). However, scientific evidence for this mechanism and clinical effectiveness through animal and human studies are still scarce. 

In a recent study, in an immunocompetent rat model of TMJ OA, MSC exosomes were observed to play an important role in the regulation of inflammatory responses, healing of condylar cartilage and subchondral bone, and pain behavior [97]. In addition, MSC exosomes are effective in treating critical-sized cartilage defects in an immunocompetent rat model [98]. In particular, MSC exosomes in TMJ OA can achieve a therapeutic effect by controlling the inflammatory response; i.e., exosome-mediated recovery in TMJ OA can be obtained while reducing the inflammatory response and is characterized by early suppression of pain [99]. Subsequent matrix expression, sustained proliferation, and gradual improvement of subchondral bone structure can lead to overall joint recovery and regeneration. Recovery may be achieved by injecting a therapeutic agent containing MSC exosomes into the damaged cartilage area (Figure 9).

Another method is to use chondrocyte cultures. Adenosine activation of AKT, ERK, and AMPK signals is obtained by adding exosome-mediated joint repair, which may increase some chondrocyte activity. Adenosine activation of AKT, ERK, and AMPK signaling can be combined with exosome-mediated repair, which increases chondrocyte activity. MSC exosomes have the ability to enhance s-GAG synthesis, which is impeded by IL-1β, and also inhibit IL-1β-induced nitric oxide and MMP13 production. The effects of these MSC exosomes can be partially inhibited by AKT, ERK, and AMPK phosphorylation and activation of adenosine receptors. Together, our investigations so far have shown that MSC exosomes work through several cellular processes that restore the matrix layer and overall homeostasis. Previous studies demonstrate the potential of exosome-based therapies available as a cell-free approach to treat TMJ pain and degeneration. Exosomes could induce intercellular communication of recipient and parent cells through their cargo at the site of TMJ OA [66,67]. This method would not only be safe and effective in inducing cartilage and bone regeneration but will also reduce pain intensity and restore mandibular function. Thus, we suggest that MSC exosomes could facilitate recovery and regeneration in TMJ OA through a sophisticated mechanism of action. However, there is only limited evidence proving effectiveness of MSC exosomes in TMJ regeneration. Further research is needed to ameliorate our view.

## 7. Conclusions 

Currently, MSC exosomes are widely accepted as major therapeutic agents derived from MSC secretions. MSC exosomes are considered sufficient to promote many of the reported therapeutic effects of MSCs. Research on the benefits of MSC exosome therapy for degenerative diseases, such as systemic OA and TMJ OA, is ongoing. Damaged articular cartilage is at the center of the pathogenesis of OA. OA symptoms can be alleviated by repairing and regenerating with MSC exosomes. Whether naturally occurring or engineered, MSC exosomes can provide therapeutic benefits. Treatment with MSC exosomes produced several positive results. However, most of the studies that yielded these results were conducted using animal models, not human models. While OA is classified as a chronic disease, there are limitations to conducting experiments in animal models with acute injuries. TMJ OA is also characterized by low levels of chronic inflammatory conditions and is a multifactorial chronic degenerative disease. To demonstrate the potential benefits of MSC exosomes towards TMJ OA, further research is needed on the pathogenesis of this degenerative disease, the potential of various MSC sources, appropriate therapeutic concentrations of MSC exosomes, and accompanying treatment regimens. Studies using rat models provide an easy-to-access safe paradigm for traditional cell-based MSC therapies. However, to conduct clinical trials in humans, MSC exosomes in large animal models should be used to repair and regenerate cartilage lesions to validate the safety and efficacy of cell-free exosome therapy. The application of MSC exosomes to treat TMJ OA can be rationalized by scientific evidence, but the ultimate goal of clinical application should be validated in the appropriate animal and human model. Subsequently, rigorous evaluation in properly controlled clinical trials is required. In conclusion, the challenge remains, but the use of exosomes in the treatment of TMJ OA has great potential with regards to their mechanism of action and therapeutic efficacy. These small vesicles can have a significant impact on the future of TMJ OA treatment.

## Figures and Tables

**Figure 1 ijms-21-01541-f001:**
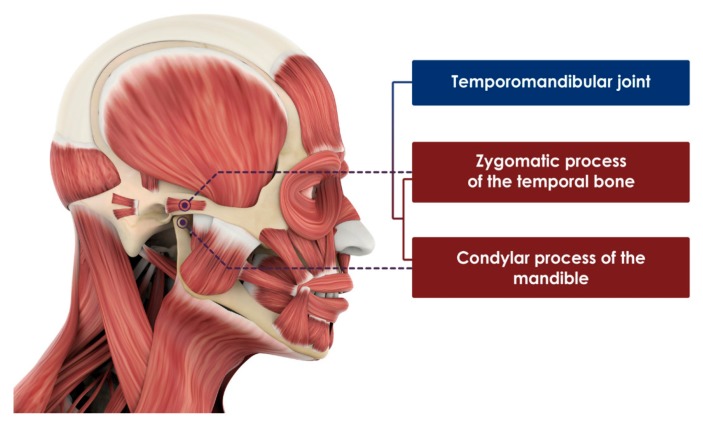
Anatomical structure of temporomandibular joint and surrounding tissues.

**Figure 2 ijms-21-01541-f002:**
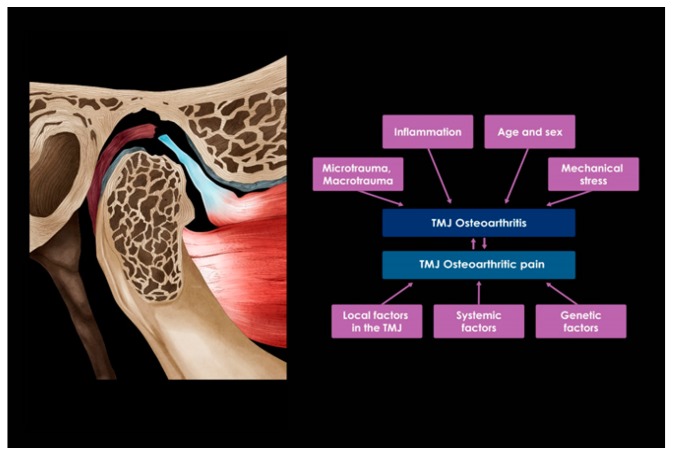
Causes of osteoarthritis on the temporomandibular joint.

**Figure 3 ijms-21-01541-f003:**
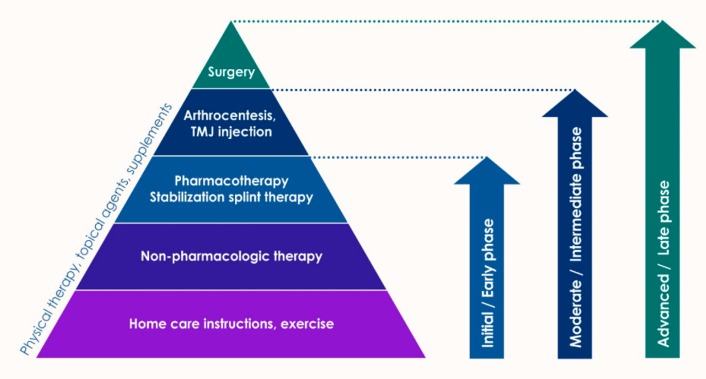
Clinical and classical treatment for osteoarthritis on the temporomandibular joint.

**Figure 4 ijms-21-01541-f004:**
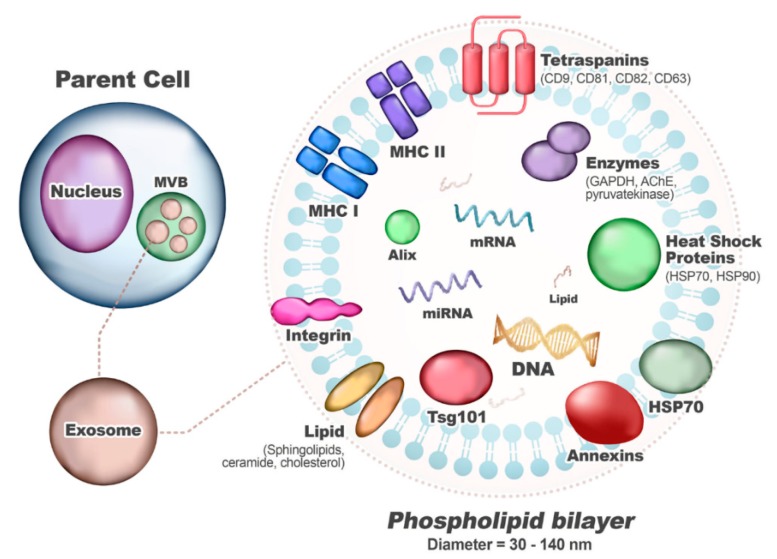
Typical content and structure of exosomes. Abbreviations: MVB, multi-vesicular bodies; MHC, major histocompatibility complex; CD, Cluster of differentiation; GAPDH, glyceraldehyde-3-phosphate dehydrogenase; HSP, heat shock protein; AChE, acetylcholinesterase; TSG101, tumor susceptibility gene 101.

**Figure 5 ijms-21-01541-f005:**
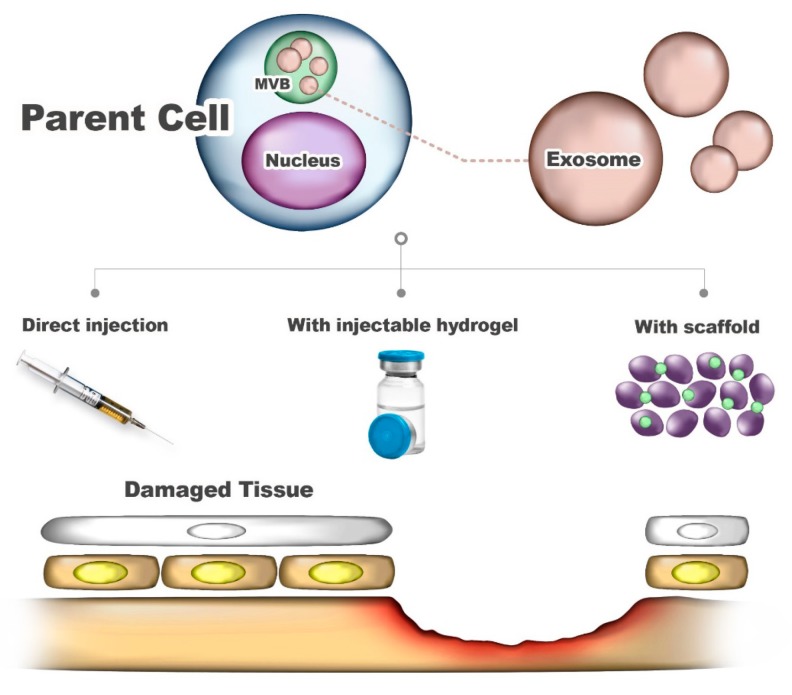
Application of exosomes to damaged tissue.

**Figure 6 ijms-21-01541-f006:**
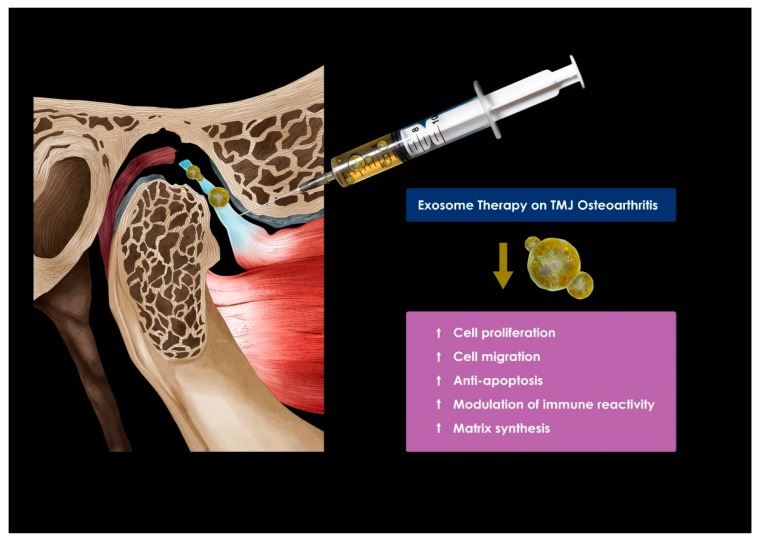
A new approach to temporomandibular joint (TMJ) osteoarthritis using exosomes.

**Figure 7 ijms-21-01541-f007:**
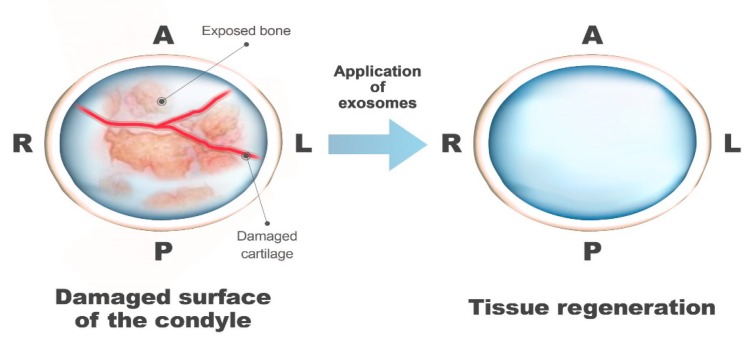
Applying exosomes to the damaged surface of the condyle. A: anterior, P: posterior, R: right, L: left.

**Figure 8 ijms-21-01541-f008:**
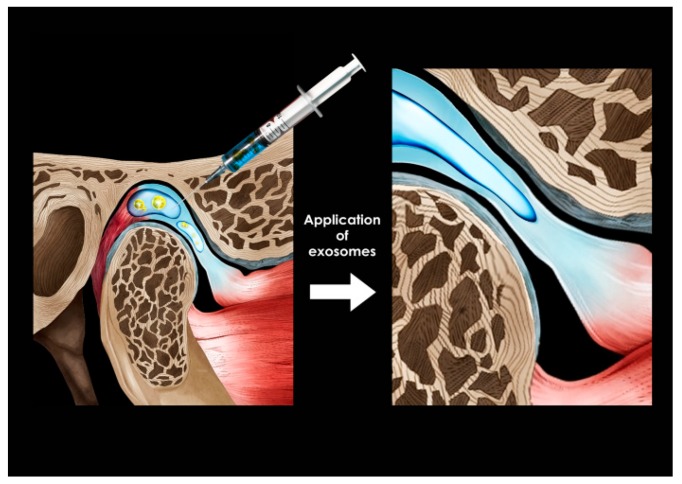
The potential role of exosomes in recovering cell numbers on degenerative disk.

**Figure 9 ijms-21-01541-f009:**
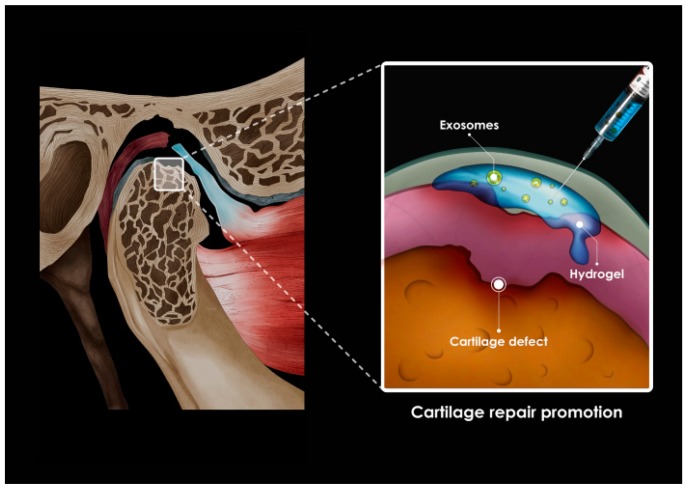
Exosome application to the destructed cartilage and subchondral bone.

**Table 1 ijms-21-01541-t001:** Advantages and disadvantages of MSCs and MSC exosomes for clinical applications [71,72,73,74,75].

	Advantages	Disadvantages
**Mesenchymal stem cell (MSC)**	Easy to isolate and obtain	Risk of teratoma formation after transplantation
	Highly proliferative	Limited number of cells
	Multilineal differentiation	Risk of potentially transmit infection
	Minimal risk of immune problems	Risk of potentially transmit genetic diseases
	In some cases, free from ethical issues	Ethical and political issues
	Accumulated experimental and clinical results (relatively long time)	Uncertainty of the related regenerative mechanism
**MSC exosomes**	Targeting efficiency through specific proteins in the exosome membranes and natural homing ability	No recommended isolation protocol
	Excellent immune-compatibility and non-cytotoxic	No standard manufacturing methods
	Low risk of teratoma formation	Rapid clearance from blood after administration (in vivo)
	Relatively free against ethical issues	Limited and insufficient research on exosome-based therapeutics
	Good delivery vehicle for both hydrophobic and hydrophilic drugs	Difficulty in isolation and purification of exosomes with specific bioactive molecules
	Stable upon freezing and thawing (compared with cells)	Lack of techniques and methodology to strictly quantify the molecular and physical aspects of exosomes

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
