# Peer review of "Emerging Potential of Exosomes in Regenerative Medicine for Temporomandibular Joint Osteoarthritis"

_ijms, 2020, doi:10.3390/ijms21041541_

Round 1

Reviewer 1 Report

Author tried to review the literature on exosomes applications for Temporomandibular Joint Osteoarthritis but he needs to rewrite this whole papers, as most of the article contains very basic knowledge about Temporomandibular Joint Osteoarthritis conventional treatments. Author need to do extensive review on current applied research on exosomes and their applications for Temporomandibular Joint Osteoarthritis. He needs to includes 3 to 4 tables and 5 to 6 original figures. Current figures very generic and not original. 

Author Response

To Reviewer 1,   In your sincere advice, the authors would like to express our sincere gratitude. As for your advice, we have discussed several times over the last eight days and have tried our best to incorporate your comments into our review. You pointed out that the content or organization of our paper is not new, but we reviewed 100 papers comprehensively on temporomandibular joints and disorders, the causes of osteoarthritis of temporomandibular joints, classical treatments, and new treatment strategies using exosomes. By organizing this intensive paper and presenting our expert opinions related to it, I think it will be easy for those interested in this topic to grasp the content. Our authors hope and expect this. In addition, the structure and flow of these papers has originality, which has been determined after numerous discussions by the authors. We didn't add a table, but we added Figure 4 to help readers understand it. Once again, thank you for your valuable advice.
Note) We have modified the English words or English grammar as a whole, and marked the additions in red. We sincerely hope you confirm this carefully.
  Thank you.

Reviewer 2 Report

The authors present a detailed review of TMJ injury and the potential for the use of exosomes as a therapeutic treatment. Overall the manuscript is well written and there is a logical progression of topics throughout the text. There are however a couple of points that should be addressed prior to publication.

The authors appear to rely heavily on the works of three other groups in formulating their review. MSC exosome as a cell-free MSC therapy for cartilage regeneration: Implications for osteoarthritis treatment https://doi.org/10.1016/j.semcdb.2016.11.008, TMJ Disorders: Future Innovations in Diagnostics and Therapeutics Journal of Dental Education August 2008, 72 (8) 930-947, and Current Understanding of Pathogenesis and Treatment of TMJ Osteoarthritis. https://doi.org/10.1177/0022034515574770. Given the age of these publications, the authors may want to consider some more recent publications particularly with regard to the exosome section which is somewhat less detailed than the TMJ section. The authors provide a very comprehensive review about TMJ injury and disease progression, but the section on exosomes is split with focus given to both MSCs and exosomes. I feel given the title of the manuscript more emphasis should be given to describing what exosomes are and their potential cargo and mechanism of action rather than comparing their effects to MSCs. Figure 1 doesn't really represent a diagram of the TMJ and should be revised to actually show the joint itself.

Author Response

Dear reviewer 2,   In your sincere advice, the authors would like to express our sincere gratitude. For your advice, we have discussed several times in the last eight days and have tried to incorporate your comments in our review paper into our research. We have carefully edited all of the points you pointed out, especially regarding the reference. Moreover, we have added the latest references to our review paper. The revised portion is marked in red in the paper.
As a result, we added seven new papers to our review study, and our papers had a total of 100 papers as references.
In addition, the authors agree with your opinion to focus more on content and flow on exosomes than on MSC. We not only deleted the contents of MSC, but left only the main parts, and reinforced the contents of MSC exosomes in the paper. In addition, figure 1 has been modified to clarify the purpose of including it in the text. The location of the temporal bone and mandibular condyle is clearly marked and the inclusion relationship with the 'temporomandibular joint' is also indicated. Once again, thank you for your valuable advice.
Note) We have modified the English words or English grammar as a whole, and marked the additions in red. We sincerely hope you confirm this carefully.
Thank you.   Best regards, 

Reviewer 3 Report

In the present review, Lee and colleagues attempt to organize and summarize the most recent literature on temporomandibular joint osteoarthritis (TMJ OA), its causes, current therapy and new potential regenerative approaches. In this regard, the authors focus on the role of Mesenchymal Stem Cell-derived exosomes as novel resources for TMJ cartilage and bone regeneration.

Overall, the review looks comprehensive and it encompasses most of the notions that concern the intended topics. Moreover, sections appear to be well organized, making it easy for the reader to follow. However, some minor revisions should be addressed (listed below), before considering the review ready for publication.

Minor revisions:

Figure 1 is supposed to show the anatomy of TMJ with the help of colour codes, however the different colours are not visible: please revise; The sentence in lines 211-2 ("precise mechanism..recently") should include a reference about the recent study the authors are referring about; Most references cite other reviews, some of them being quite old, however it would seem more appropriate to cite the original articles whenever possible, or to refer to more recent data. For instance reference #7 refers to women-specific studies, but it is from 2001: aren't there more recent studies?

Author Response

To Reviewer 3, In your sincere advice, the authors would like to express our sincere gratitude. For your advice, we have discussed several times in the last eight days and have tried to incorporate your comments in our review paper into our research. 1. We have modified Figure 1 more clearly.
2. We have inserted a reference to the part you pointed at ("precise mechanism..recently").
3. We have found more recent studies and data to support our claim or explanation, and we have added seven more recent articles to the text. As a result, a total of 100 papers were reviewed. Sincerely thank you. Based on your advice, I think our quality of the paper has improved further.
Once again, thank you for your valuable advice.
We have modified the English words or English grammar as a whole, and marked the additions in red. We sincerely hope you confirm this carefully.
Thank you.     Best regards, 

Round 2

Reviewer 1 Report

Authors tried to improve the paper but still it is not sufficient to call it as review papers, Kindly add two tables by extracting information of papers you mentioned 100 and compare the positive and negative impacts of old and modern techniques based on exosomes while include a second tables based on clinical trials done so far by using exosomes. Addition of figure 4 is good improvements, draw similar 4 to 5 more figures to depict different aspects of exosomes and its applications in regenerative therapy. 

Author Response

To Reviewer 1,

In your sincere and constructive advice, the authors would like to express our sincere gratitude.

As for your advice, we have discussed several times and have tried our best to incorporate your comments into our review paper.

You asked us to improve our paper. In particular, you advised us to add tables and figures. First of all, we tried to pinpoint the meaning of what you pointed out. In addition, we have decided to reflect further discussions and follow your recommendations. Specifically, we added four figures to satisfy your point. We determined that this would help readers of the journal understand the content of 100 papers and visually understand the content. And, we added Table 1. In Table 1, we summarize the advantages and disadvantages of clinical application and treatment based on MSC exosomes, and how they compare to the methods using MSC.

As you know, our review paper encompasses 100 papers published previously. Through this review, we have intensively summarized various aspects of the potential of exosomes in TMJ, TMJ-OA, and TMJ-OA treatment, and presented our opinions.

In other words, we do not intend to provide detailed descriptions of individual items such as the appropriate concentration, separation method, and mass production method in OA treatment using exosome.

Rather than describing the individual items presented above, we focus on presenting fresh perspectives on new therapies using exosomes. We swear, indeed, sincerely that we have done our best to understand your needs and incorporate them into our papers.

Once again, thank you for your valuable advice.

Note) We sincerely hope you carefully read the additional revisions (marked in red).

Thank you.

Best regards,

Reviewer 2 Report

The authors have considered the reviewer's comments and have revised the manuscript thoughtfully and have produced a more focused and more detailed review of the TMJ and potential exosome treatments. I am happy with the revisions made to the manuscript.

Author Response

To Reviewer 2,

In your sincere advice, the authors would like to express our sincere gratitude. With your understanding and praise of the part of the paper we modified, we were really happy.

In addition, we added four figures and 1 table. We determined that this would help readers of the journal understand the content of 100 papers and visually understand the content. In Table 1, we summarize the advantages and disadvantages of clinical application and treatment based on MSC exosomes, and how they compare to the methods using MSC.

Once again, thank you for your valuable advice.

Note) We sincerely hope you carefully read the additional revisions (marked in red).

Thank you.

Best regards,

Round 3

Reviewer 1 Report

N/A